# Nutritional Intervention in Cushing’s Disease: The Ketogenic Diet’s Effects on Metabolic Comorbidities and Adrenal Steroids

**DOI:** 10.3390/nu15214647

**Published:** 2023-11-02

**Authors:** Valentina Guarnotta, Roberta Amodei, Francesca Di Gaudio, Carla Giordano

**Affiliations:** 1Section of Endocrinology and Diabetology, Health Promotion, Department of Health Promotion, Mother and Child Care, Internal Medicine and Medical Specialties “G. D’Alessandro”, PROMISE, University of Palermo, 90127 Palermo, Italy; valentina.guarnotta@unipa.it (V.G.); roberta.amodei@gmail.com (R.A.); 2Department of Health Promotion, Mother and Child Care, Internal Medicine and Medical Specialties, University of Palermo, CQRC (Quality Control and Chemical Risk) Hospital Company, Hospitals Reunited Villa Sofia Cervello, 90146 Palermo, Italy; francesca.digaudio@unipa.it; 3Institute of Translational Pharmacology (IFT), National Research Council of Italy (CNR), 90146 Palermo, Italy

**Keywords:** very low-calorie ketogenic diet, cortisone, diabetes mellitus, obesity

## Abstract

Background: a very low-calorie ketogenic diet (VLCKD) is associated with improvement of metabolic and cardiovascular disorders. We aimed to evaluate the effects of a VLCKD in patients with Cushing’s disease (CD) as adjunctive therapy to treatment for the primary disease. Methods: we evaluated clinical, hormonal and metabolic parameters in 15 patients with CD and 15 controls at baseline after 1 week and 3 weeks of VLCKD and, further, after 2 weeks of a low-carbohydrate ketogenic diet (LCKD). Results: after 5 weeks of diet, a significant decrease in BMI (*p* = 0.002), waist circumference (WC) (*p* = 0.024), systolic blood pressure (*p* = 0.015), diastolic blood pressure (*p* = 0.005), ACTH (*p* = 0.026), cortisone (*p* = 0.025), total cholesterol (*p* = 0.006), LDL cholesterol (*p* = 0.017), triglycerides (*p* = 0.016) and alkaline phosphatase (*p* = 0.008) and a significant increase in HDL cholesterol (*p* = 0.017), vitamin D (*p* = 0.015) and oral disposition index (oDI) (*p* = 0.004) was observed in the CD patients. A significant decrease in BMI (*p* = 0.003), WC (*p* = 0.002), systolic blood pressure (*p* = 0.025), diastolic (*p* = 0.007) blood pressure and total cholesterol (*p* = 0.026) and an increase in HDL cholesterol (*p* = 0.001) and oDI (*p* < 0.001) was observed in controls. Conclusions: the current study confirms that a ketogenic diet is effective in improving metabolic disorders in CD and shows that a nutritional approach may be combined with conventional CD therapy in order to improve metabolic and cardiovascular comorbidities.

## 1. Introduction

Cushing’s syndrome (CS) is a clinical condition characterized by an excess of glucocorticoids. It can be exogenous, resulting from a chronic intake of synthetic corticosteroids, or endogenous, resulting from hyperproduction of cortisol or adrenocorticotropic hormone (ACTH). The endogenous hypercortisolism can be ACTH-dependent due to a pituitary ACTH hypersecretion defined as Cushing’s disease (CD) or to extrapituitary hyperproduction of ACTH, defined as ectopic CS or ACTH-independent CS due to adrenal cortisol hypersecretion [1].

CS is associated with increased mortality, compared to the general population, due to the presence of several comorbidities, including visceral obesity, diabetes mellitus, arterial hypertension, cardiovascular disease, osteoporosis and musculoskeletal disorders including myopathy, dyslipidaemia, infections and neuropsychiatric and reproductive disorders [2,3].

With regard to CD, the first therapeutic approach consists of pituitary surgery. In cases of contraindications for surgery, the patients’ refusal, or disease recurrence, medical therapy, including inhibiting pituitary- or adrenal-directed drugs, radiotherapy or bilateral adrenalectomy, should be recommended [4].

Generally, the treatment of CD is associated with improvement of comorbidities, even though some of them can also persist in the remission phase [5].

The low-carbohydrate diet approach and, notably, the very low-calorie ketogenic diet (VLCKD) have several therapeutic applications, improving many metabolic disorders, including diabetes mellitus, obesity, arterial hypertension, insulin resistance and dyslipidaemia, as strongly supported by evidence [6,7,8,9]. All of these metabolic disorders are present in patients with CD.

The effects of a VLCKD on cortisol levels have not been yet clearly elucidated [10]. Indeed, it is widely known that meal macronutrients have a strong influence on cortisol concentrations, inducing an increase or decrease in them, as reported in many studies [11,12,13,14,15,16,17,18].

Very few studies are currently available on the usefulness of a nutritional approach for the management of patients with CS. Recently, Dugandzic et al. described the case of a patient with CD who experienced beneficial effects on metabolic complications, correlated with CD, from following a low carb diet [19].

VLCKD is a subtype of a low-carbohydrate diet, which provides a low daily caloric intake (less than 800 kcal/day), low carbohydrate intake (<50 g/day) and normoproteic (1–1.5 g of protein/kg of ideal body weight) contents [5,8,20]. A VLCKD is characterized by about 13% carbohydrates, 44% protein and 43% fats.

A VLCKD induces ketogenesis, which takes advantage of ketone bodies as an energy source derived from fatty acids (Figure 1).

Ketone bodies are used by many tissues, including those of the heart, kidney, skeletal muscle and central nervous system. In physiological conditions, Acetyl-CoA can be combined with oxaloacetate that is obtained by glycolytic processes. A VLCKD is associated with a slowed glycolysis, which results in the use of oxaloacetate for neoglucogenesis, while the Acetyl-CoA obtained from the beta oxidation of fatty acids is used for the production of ketone bodies. The VLCKD diet is the model with the greatest availability of Acetyl-CoA [20,21]. Generally, a VLCKD includes six phases. In Phase 1, patients are educated to eat high-biological-value protein preparations five times a day and vegetables with a low glycemic index. Meal preparations contain 18 g of proteins, 4 g of carbohydrates and 3 g of fats. In Phase 2, a portion of natural proteins, including meat/egg/fish, can be introduced at lunch or dinner combined with a protein preparation. In Phase 3, a second portion of natural protein is added in place of the protein preparation. Phases 2 and 3 correspond to the LCKD protocol characterized by 800–1200 Kcal/day, with 13% carbohydrate, 29% proteins and 58% fats.

After, a low-carbohydrate diet with a daily calorie intake ranging from 1200 to 1500 Kcal/day is started. Carbohydrates are gradually reintroduced. First, foods with a lower glycemic index are introduced, including fruit and milk products (Phase 4), followed by moderate glycemic index food such as legumes (Phase 5) and high glycemic index ones (bread, pasta and cereals—Phase 6). At the end of Phases 4–6, a maintenance diet of approximately 1500–2000 Kcal/day is recommended [7,21].

The primary objective of the current study was to assess the effects of a 3-week VLCKD and a 2-week LCKD, as adjunctive treatment to medical therapy for CD, on salivary and serum cortisol, adrenal steroids, ACTH and serum cortisol after a 1 mg dexamethasone suppression test (DST) and urinary free cortisol in 15 patients with CD and 15 controls.

The secondary objective of the study was to evaluate anthropometric, clinical and metabolic parameters before and after a 3-week VLCK and a 2-week LCKD in both patients with CD and controls.

## 2. Materials and Methods

### 2.1. Study Population

We prospectively enrolled 15 patients with active CD (11 women (75%), 4 men (25%) (mean age 47.2 ± 10.6 years; mean BMI 35.7 ± 4.5 kg/m^2^)) and 15 controls, age-matched, BMI-matched and sex-matched, who were consecutively referred to the Division of Endocrinology of Palermo University from January 2022 to June 2023.

CD was diagnosed as recommended by the international clinical practice guidelines and consensus statement [1]. The control group was recruited in parallel to the CD group.

Patients with CD were all pharmacologically treated for the primary disease. Ten were treated with a pituitary-directed drug, pasireotide, and five with an adrenal-directed drug, metyrapone. Medical treatment for CD was maintained for the entire duration of the study.

After enrolment, patients with CD and controls were instructed to follow a nutrition plan of VLCKD for 3 weeks followed by 2 weeks of LCKD. Inclusion criteria were age 18 years or older and active CD on medical treatment for the group of patients with CD; age 18 years or older and absence of any known disease for the control group.

Exclusion criteria were the following: previous pituitary surgery or radiotherapy within 6 months prior to study entry for patients with CD; history or presence of epilepsy; mental disease rendering patients unable to understand the nature, scope and possible consequences of the study and/or evidence of an uncooperative attitude; pregnancy or breastfeeding; underweight (BMI < 18.5 kg/m^2^); adrenal CS and ectopic CS.

Among patients with CD, 9 out of 15 had diabetes mellitus, 6 were on metformin treatment and 3 were on GLP-1 receptor agonists which were suspended 1 week before the start of VLCKD protocol. In total, 7 out of 15 had arterial hypertension and were pharmacologically treated with ACE inhibitors. In total, 10 out of 15 had dyslipidaemia and were treated with statins.

Patients with CD and controls were instructed to follow a 3-week VLCKD according to a specific plan consisting of four replacement meals (by New Penta, Cuneo, Italy), subdivided into breakfast, lunch, snack and dinner, and two portions of low glycemic index vegetables at lunch and dinner, for a total amount of 670 Kcal/day.

After, a 2-week LCKD plan was prescribed consisting of three replacement meals, one conventional protein meal and two portions of low glycemic index vegetables for lunch and dinner for a total of about 820 Kcal/day. During the entire study period, the patients took Pentacal plus, a multi-vitamin and multi-mineral supplement containing magnesium (187 mg), potassium (1000 mg), vitamin C (60 mg), vitamin E (9 mg), selenium (41 mcg), vitamin A (0.6 mg), vitamin B6 (1.5 mg), vitamin B1 (1.05 mg), vitamin B2 (1.2 mg), folic acid (200 mcg) and vitamin B12 (1.87 mcg).

Before starting the ketogenic protocol, the study participants were educated about the transient disturbances that could occur in the initial phase of VLCKD: acetonemic breath, headache, hunger, constipation/diarrhea, cramps, nausea, fatigue, dizziness, menstrual cycle changes.

The study protocol was approved by the ethics committee of the Policlinico Paolo Giaccone under number 03/2022.

At the time of enrolment, each patient provided their written consent to freely participate in the experimental study.

### 2.2. Study Design

Both in patients with CD and in controls, at baseline (time 0) and after 1 week and 3 weeks of VLCKD and a further 2 weeks of LCKD (after a total time of 5 weeks), we evaluated clinical parameters (BMI, waist circumference and systolic and diastolic blood pressure), hormonal parameters (urinary free cortisol (UFC) on 24 h urine collection, late night salivary cortisol at 11 pm, serum cortisol at 8 am, serum ACTH at 8 am, serum cortisone, serum 17hydroxyprogesterone (17OHP), androstenedione and DHEAS) and metabolic parameters (fasting glucose and insulin; total, HDL and LDL cholesterol; triglycerides, glutamic oxaloacetate transaminase (GOT); glutamic piruvate transaminase (GPT); alkaline phosphatase; gammaGT; blood count; creatinine; sodium; potassium; calcium; phosphorus; parathyroid hormone; vitamin D and C-reactive protein). Only in patients with CD did we evaluate serum cortisol the next day, after 1 mg DST at baseline and after 1, 3 and 5 weeks.

In addition, we measured glycated hemoglobin (HbA1c) only at baseline and we performed an oral glucose tolerance test (OGTT) with evaluation of serum glucose and insulin at baseline and after (30–60–90–120 min) administration of 75 g of glucose. We further calculated Matsuda insulin sensitivity index (ISI-Matsuda) [10,000/glucose (mg/dL) × insulin (mU/mL) × mean blood glucose × mean insulin], the oDI (oral disposition index) [(ΔInsulin0–30/ΔGlucose0–30) × (1/fasting insulin)] and the area under the curves of insulin (AUC2-h insulinemia) and glucose (AUC2-h glycemia) at baseline and after 5 weeks, both in patients with CD and controls.

### 2.3. Assays

Height and body weight were obtained at the outpatient’s clinic; patients were weighed clothed without shoes. The waist circumference was defined as the minimal abdominal circumference located midway between the lower rib margin and the iliac crest, using a flexible tape measure and maintaining close contact with the skin without compression of underlying tissues.

Blood chemistry parameters were collected after overnight fasting. Blood glucose, insulin, HbA1c, total cholesterol, HDL cholesterol, LDL cholesterol, triglycerides, GOT, GPT, alkaline phosphatase, gamma GT, blood count, creatinine, sodium, potassium, calcium, phosphorus, parathyroid hormone, vitamin D, ACTH and c-reactive protein were calculated by standard methods (Modular P800, Roche, Milan, Italy).

UFC, serum cortisol and adrenal steroids (cortisone, 17OHP, androstenedione and DHEAS) and salivary cortisol were determined by high-performance liquid chromatography–mass spectrometry (Agilent HPLC series 1200), an Agilent 6430 triple quadrupole mass spectrometer equipped with an electrospray ionization source, operating in positive ion mode (Agilent Technologies, Palo Alto, CA, USA), as previously reported [22].

### 2.4. Statistical Analysis

IBM SPSS Statistics version 19.0 (IBM Corp., Armonk, NY, USA) was used for data analysis. The normality of quantitative variables was tested with the Shapiro–Wilk test. Data were presented as mean ± SD for continuous variables. Rates and proportions were calculated for categorical variables. The differences between paired continuous variables (CD vs. controls) were analyzed using one-way ANOVA. A *p*-value of 0.05 was considered statistically significant.

## 3. Results

All patients included in the study completed the protocol. Some adverse events were registered: 10 patients with CD and 11 controls experienced acetonemic breath and headache; 3 patients with CD and 2 controls experienced hunger; 9 patients with CD and 10 controls experienced constipation. Patients with CD at baseline had significantly higher WC (*p* = 0.035), ACTH (*p* = 0.043), UFC (*p* < 0.001), salivary cortisol (*p* < 0.001), cortisol after 1 mg of DST (*p* = 0.002), androstenedione (*p* < 0.001), DHEAS (*p* = 0.008), HbA1c (*p* = 0.045), AUC2h glycemia (*p* = 0.022) and AUC2h insulinemia (*p* = 0.040) and lower levels of vitamin D (*p* = 0.033), oDI (*p* < 0.001) and ISI-Matsuda (*p* = 0.049) compared to controls (Table 1).

In patients with CD, we observed a significant decrease in BMI (*p* = 0.002), WC (*p* = 0.024), systolic (*p* = 0.015) and diastolic blood pressure (*p* = 0.005), ACTH (*p* = 0.026), cortisone (*p* = 0.025), total (*p* = 0.006) and LDL cholesterol (*p* = 0.017), triglycerides (*p* = 0.016), alkaline phosphatase (*p* = 0.008) and a significant increase in HDL cholesterol (*p* = 0.017), vitamin D (*p* = 0.015) and oDI (*p* = 0.004) during the 5 weeks of observation (Table 2 and Figure 2).

A significant decrease in BMI (*p* = 0.003), WC (*p* = 0.002), systolic blood pressure (*p* = 0.025), diastolic (*p* = 0.007) blood pressure and total cholesterol (*p* = 0.026) and increase in HDL cholesterol (*p* = 0.001) and oDI (*p* < 0.001) was observed in controls (Table 3 and Figure 3).

Analyzing in detail patients with CD, after 5 weeks of ketogenic diet, we observed a decrease in ACTH in 12 out of 15 patients, a decrease in UFC in 10 out of 15 patients, a decrease in salivary cortisol in 6 out of 15 and a decrease in cortisol after 1 mg of DST in 4 out of 15 patients (Figure 3). By contrast, in controls, we observed a decrease in ACTH in 12 out of 15 patients, a decrease in UFC in 6 out of 15, a decrease in cortisol after 1 mg of DST in 4 out of 15 and a decrease in serum cortisol in 5 out of 15 after 5 weeks of diet.

## 4. Discussion

The current study shows that a 3-week VLCKD followed by a 2-week LCKD is associated with an improvement in clinical, lipid and insulin sensitivity parameters in patients with CD and controls.

We did not observe any changes in serum and urinary cortisol levels and ACTH levels, while a significant increase in cortisone serum levels was observed in patients with CD. We also observed an increase in vitamin D levels in patients with CD after 5 weeks of ketogenic diet.

We also showed that patients with CS had lower vitamin D levels compared to healthy controls, as recently reported [9], with significant improvement after a VLCKD. Recently, Perticone et al. reported an improvement of vitamin D values in patients who followed a ketogenic diet protocol, due to significant weight loss [23].

The relationship between fasting and caloric restriction and cortisol is complex and interesting.

Although some studies reported that caloric diet restriction is associated with an increase in serum and salivary cortisol and in UFC levels and with inadequate suppression of cortisol after a low dose of DST [12,13,17,24], this aspect was not clearly ascertained [24,25].

An interesting role in obesity is played by the enzyme 11-β-hydroxysteroid dehydrogenase type 1 (11β-HSD1), which is involved in conversion of inactive cortisone into active cortisol; it was observed that this activity could change after weight loss [25]. A study conducted on 24 obese men showed an increased cortisol production rate and increased free cortisol levels combined with decreased 11β-HSD1 after weight loss [26].

Stimson et al. reported that in a group of obese men, a lower carbohydrate diet increased the enzyme 11β-HSD1, able to activate cortisol, and reduced the enzymes 5-alpha and 5-beta reductase, which inactivated cortisol [27]. The regeneration of cortisol in the low-carb group was correlated with the carbohydrate ratio rather than with the number of calories.

In another study by Stimson et al., the effects of carbohydrate, high-protein and high-fat meals on cortisol levels were evaluated in eight lean men, showing that carbohydrates promote both adrenal cortisol secretion and extra-adrenal cortisol regeneration mediated by 11β-HSD1, while high-protein and high-fat meals stimulate adrenal cortisol secretion to a greater degree than extra-adrenal regeneration [28].

Other studies reported unchanged serum cortisol, cortisone and urinary steroid levels and decreased 11β-HSD1 after weight loss [29]. In agreement with all the above-mentioned findings, it may be hypothesized that the more significant the weight loss is, the more 11β-HSD1 activity is decreased, as also suggested by studies conducted on patients undergoing bariatric surgery [30,31].

Interestingly, the effects of fasting, a very low-calorie diet (VLCD) and a low-calorie diet (LCD) on serum cortisol levels were evaluated by a metanalysis [32]. This meta-analysis excluded studies based on salivary or urinary cortisol levels in order to prevent heterogeneity of the studies. Short-term calorie restriction was reported to be associated with an increase in cortisol values, while a VLCD and LCD had no long-term effects on serum cortisol values and were less stressful than fasting. Further, carbohydrate restriction decreased circulating insulin levels, leading to extra-adrenal cortisol synthesis [32].

In the current study, we did not observe any change in serum, urinary and salivary cortisol both in patients with CD and controls, in line with the findings reported by Nakamura et al. In contrast, our findings are different from those reported by Stimson et al. We could hypothesize that a chronic intake of high fat and normoproteic meals could act differently on cortisol levels over a long-term period of observation. Indeed, although Stimson et al. reported a significant increase in cortisol levels after high-protein and high-fat meals, the time of observation was limited to a single meal, and it is difficult to speculate whether the cortisol response could change after chronic assumption of high-protein and high-fat meals. In addition, in the study conducted by Stimson et al., meals were composed of single macronutrients. A combination of macronutrients, even with different percentages, could impact differently on the immediate circulating concentrations of cortisol levels. This hypothesis could also be supported by the evidence that a VLCKD and LCKD are always associated with weight loss; this would be discordant with a cortisol rise, which would be associated with a weight increase.

In the current study, we also evaluated patients with CD regarding pharmacological treatment for the primary disease. We could speculate that cortisol-lowering drugs could impact on the results of a lack of change on serum, salivary and urinary cortisol values. However, we also evaluated a control group of healthy people and did not observe significant changes in cortisol values, supporting the hypothesis that a ketogenic diet does not have a relevant effect on cortisol values. However, larger studies focused on the effects of a VLCKD on cortisol concentrations are required.

Nutritional intervention may be a mainstay for the treatment of CD that is able to improve metabolic complications of CD when combined with the treatment of the primary disease. These improvements may occur independently of biochemical hypercortisolism control.

Indeed, a VLCKD reduces visceral fat and decreases appetite, resulting in weight loss [33]. It reduces lipogenesis, increases lipolysis [33,34] and decreases glucose and insulin levels, improving insulin sensitivity and glycemic control in patients with type 2 diabetes mellitus and gonadal function in women with PCOS [5,35,36,37]. In addition, the natriuretic effect of ketone bodies results in a reduction in blood pressure values [38]. Controlled ketogenesis is one of the mechanisms of action of inhibitors of renal sodium-glucose co-transporter type 2 (SGLT2-inhibitors), which, beyond causing natriuresis and glycosuria, promote a shift towards ketogenesis resulting in a 38% decrease in cardiovascular mortality [39]. The cardiovascular beneficial effects induced by SGLT2-inhibitors may be related to the inhibitory action of ketone bodies on the sympathetic nervous system [40].

Further, a VLCKD improves inflammation, reducing TNF-alpha, PAI-1, IL-6, IL-8 and MCP-1, which are strongly involved in the pathophysiology of cardiovascular diseases and obesity [41]. Regarding the duration of the dietetic protocol, we investigated a short period of 5 weeks. However, studies conducted on obese patients with combined diabetes mellitus showed a valuable therapeutic effect of a VLCKD in the long-term management of obesity. So, we could hypothesize an overall duration of the ketosis phases (VLCKD and LCKD) of about 2 months, supplemented by multi-vitamin contents.

The current study shows some limitations. First, the number of patients with CD included in the study is quite low, even though CD is rare. Second, the duration of the study is brief. Third, the pharmacological treatment of CD was different among the patients. Fourth, the overall nutritional risk of an enduring implementation of this dietary protocol were not evaluated. However, the study, to our knowledge, is the first one which evaluates the usefulness of nutritional intervention in patients with CD, analyzing the effects on serum and urinary cortisol levels, beyond anthropometric, metabolic and insulin sensitivity parameters, not only in patients with CD but also in a control group.

## 5. Conclusions

The current study confirms that a VLCKD and LCKD are effective in improving metabolic disorders and shows that a nutritional approach may be combined with conventional CD therapy in order to improve metabolic and cardiovascular comorbidities. Indeed, independently of the biochemical control of hypercortisolism, metabolic comorbidities can persist. For this reason, in patients with CD, a low-carb diet, including a ketogenic one, could be started before and/or after pituitary surgery or in patients pharmacologically treated with pituitary- or adrenal-directed drugs.

## Figures and Tables

**Figure 1 nutrients-15-04647-f001:**
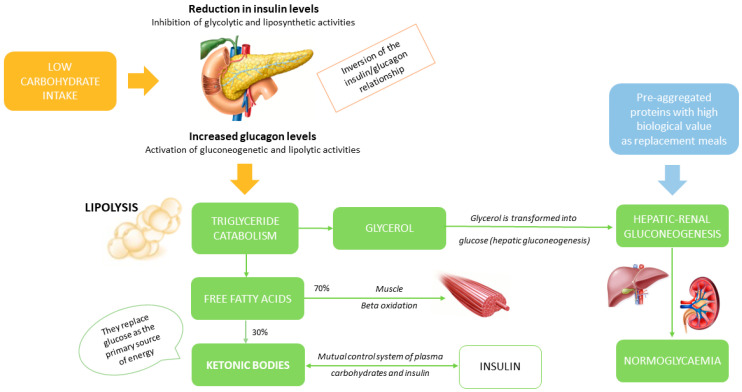
Mechanisms of ketogenesis. Reduction in exogenous glucose intake and reduction in the insulin/glucagon ratio result in reduced inhibition of lipolysis in adipose tissue. This leads to increased levels of free fatty acids circulating and to greater beta oxidation of fatty acids, with the formation of ketone bodies at the liver level and, to a lesser extent, at the renal level. Blood sugar levels are maintained in a physiological range thanks to liver function, both through mobilization from the hepatic glycogen reserve and through the process of gluconeogenesis.

**Figure 2 nutrients-15-04647-f002:**
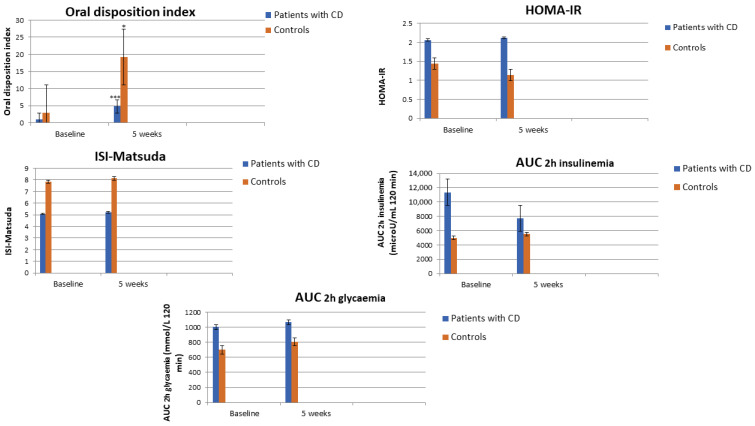
Comparison in insulin sensitivity parameters in patients with Cushing’s disease and controls at baseline and after 5 weeks of ketogenic diet (data are expressed in mean ± SD). *** *p* < 0.001; * *p* < 0.005.

**Figure 3 nutrients-15-04647-f003:**
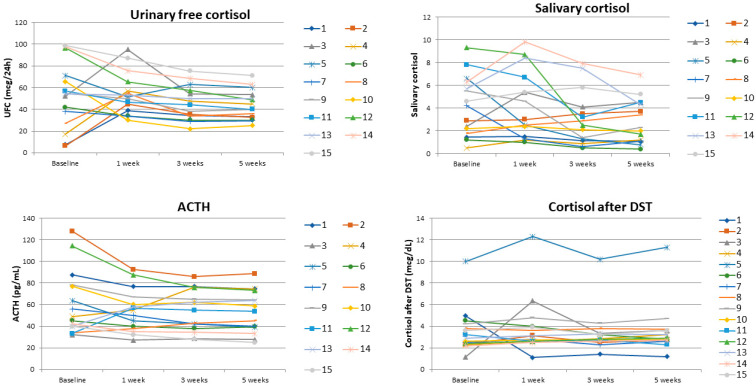
Changes in urinary free cortisol, salivary cortisol and cortisol after dexamethasone suppression test (DST) and ACTH in each patient with Cushing’s disease at baseline and after 1, 3 and 5 weeks of ketogenic diet.

**Table 1 nutrients-15-04647-t001:** General characteristics of all patients enrolled in the study.

	Patients with Cushing’s Disease (N = 15)	Controls (N = 15)	*p*
	Mean ± SD	Mean ± SD	
Clinical parameters			
Age (years)	47.2 ± 10.6	50.8 ± 11.7	0.535
BMI (Kg/m^2^)	35.7 ± 4.5	32.8 ± 2.14	0.175
Waist circumference (cm)	114.4 ± 12.7	96.6 ± 14.2	0.035
Systolic blood pressure (mmHg)	125.8 ± 15.6	122.3 ± 8.6	0.453
Diastolic blood pressure (mmHg)	88.2 ± 7.8	83.2 ± 4.8	0.057
Hormonal parameters			
Fasting serum cortisol (mcg/dL)	13.2 ± 4.11	11.2 ± 3.44	0.443
Fasting serum ACTH (ng/L)	77.8 ± 40.8	27.1 ± 19.5	0.043
Urinary free cortisol (mcg/24 h)	56.8 ± 30.9	33.4 ± 10.7	<0.001
Salivary cortisol	6.6 ± 2.76	0.76 ± 0.49	<0.001
Cortisol after low-dose suppression test (mcg/dL)	4.14 ± 3.55	0.99 ± 0.66	0.002
Androstenedione (mcg/L)	2.86 ± 2.31	0.71 ± 0.34	<0.001
DHEAS (mcg/dL)	1743 ± 1110	624.5 ± 324.5	0.008
Cortisone (mcg/L)	15.1 ± 3.9	18.6 ± 6.5	0.624
17OHP (mcg/L)	0.53 ± 0.22	0.32 ± 0.21	0.339
Metabolic parameters			
Glycemia (mmol/L)	4.91 ± 0.83	4.44 ± 0.34	0.239
Insulinemia (microU/mL)	9.48 ± 8.44	7.46 ± 3.83	0.639
HbA1c (%)	5.96 ± 0.96	5.14 ± 0.32	0.045
Total cholesterol (mmol/L)	4.96 ± 0.6	4.97 ± 0.6	0.637
HDL cholesterol (mmol/L)	1.36 ± 0.28	1.38 ± 0.53	0.958
Triglycerides (mmol/L)	1.2 ± 0.5	0.82 ± 0.37	0.116
LDL cholesterol (mmol/L)	2.63 ± 0.58	2.8 ± 0.44	0.931
GOT (U/L)	17.5 ± 5.06	22.2 ± 10.8	0.131
GPT (U/L)	22.2 ± 10.8	22.8 ± 6.96	0.919
Alkaline phosphatase (U/L)	71.7 ± 21.6	72.2 ± 19.6	0.975
GammaGT (U/L)	15.1 ± 8.01	15 ± 12.1	0.445
Creatinine (mg/dL)	0.72 ± 0.08	0.71 ± 0.13	0.854
Na (mmol/L)	140.2 ± 2.16	139.8 ± 1.78	0.758
K (mmol/L)	4.4 ± 0.42	4.2 ± 0.44	0.408
Calcium (mg/dL)	9.36 ± 0.25	9.31 ± 0.26	0.717
Phosphorus (mg/dL)	3.28 ± 0.29	3.56 ± 0.23	0.395
Vitamin D (mcg/L)	21.4 ± 5.22	25.6 ± 5.09	0.033
PTH (ng/L)	52.7 ± 17.5	43 ± 11.8	0.351
C-reactive protein (mg/L)	2.44 ± 1.95	0.79 ± 0.75	0.145
Dio	0.97 ± 0.66	2.91 ± 2.77	<0.001
AUC 2h insulinemia (uU/mL 120 min)	11,370 ± 5592	4977 ± 1636.5	0.040
AUC 2h glycemia (mmol/L 120 min)	1004.1 ± 330.3	700.6 ± 111.9	0.022
HOMA-IR	2.06 ± 1.95	1.44 ± 0.71	0.521
ISI-Matsuda	5.08 ± 3.66	7.85 ± 3.73	0.049

Abbreviations: oDI oral disposition index, AUC 2h insulinemia (area under the curve of insulin) and AUC 2h glycemia (area under the curve of glucose), PTH, parathyroid hormone.

**Table 2 nutrients-15-04647-t002:** Clinical, hormonal and metabolic parameters in patients with Cushing’s disease at baseline and after 1, 3 and 5 weeks of ketogenic diet.

	Patients with Cushing’s Disease (N = 15)				
	Baseline Mean (± SD)	Week 1 Mean (± SD)	Week 3 Mean (± SD)	Week 5 Mean (± SD)	*p*
Clinical parameters					
BMI (Kg/m^2^)	35.7 ± 6.54	34.4 ± 6.15	33.2 ± 5.91	32.3 ± 5.53	0.002
Waist circumference (cm)	114.4 ± 12.7	111.6 ± 11.1	109 ± 10.7	107.8 ± 8.16	0.024
Systolic blood pressure (mmHg)	125.8 ± 15.6	120.4 ± 11.8	118.4 ± 10.3	113.8 ± 9.6	0.015
Diastolic blood pressure (mmHg)	88.2 ± 7.8	85.4 ± 5.8	84.2 ± 6.1	82.1 ± 5.1	0.005
Hormonal parameters					
Fasting serum cortisol (mcg/dL)	11.4 ± 2.22	12.7 ± 3.33	14.2 ± 3.08	14.7 ± 2.53	0.122
Fasting serum ACTH (ng/L)	77.8 ± 40.4	60.3 ± 29.5	58.2 ± 27.3	41.8 ± 26.9	0.026
Urinary free cortisol (mcg/24 h)	56.8 ± 30.9	67.9 ± 43.3	55.6 ± 43.7	52.3 ± 37.6	0.472
Late-night salivary cortisol	6.67 ± 4.76	7.38 ± 3.32	4.76 ± 3.73	2.67 ± 2.33	0.160
Cortisol after low dose suppression test (mcg/dL)	4.14 ± 3.55	3.89 ± 3.03	3.88 ± 3.01	3.06 ± 2.85	0.139
17OHP (mcg/L)	0.53 ± 0.22	0.63 ± 0.21	0.85 ± 0.18	0.76 ± 0.14	0.897
Androstenedione (mcg/L)	2.86 ± 2.31	2.5 ± 2.11	2.36 ± 1.47	3.05 ± 1.98	0.558
DHEAS (mcg/L)	1743 ± 1110	1892.5 ± 165.6	1871.7 ± 333.5	1656.7 ± 323.5	0.125
Cortisone (mcgd/L)	15.1 ± 3.9	16.6 ± 4.01	17.8 ± 3.85	19.8 ± 4.51	0.025
Metabolic parameters					
Glycemia (mmol/L)	4.91 ± 0.83	4.88 ± 1.11	4.94 ± 0.55	4.46 ± 0.7	0.356
Insulinemia (microU/mL)	6.7 ± 4.02	5.8 ± 3.11	4.15 ± 3.33	5.63 ± 5.02	0.241
Total cholesterol (mmol/L)	4.96 ± 0.6	4.92 ± 0.83	4.16 ± 0.75	4.06 ± 0.81	0.006
HDL cholesterol (mmol/L)	1.36 ± 0.28	1.43 ± 0.28	1.46 ± 0.35	1.63 ± 0.3	0.017
Triglycerides (mmol/L)	1.2 ± 0.5	0.9 ± 0.23	0.89 ± 0.2	0.75 ± 0.15	0.040
LDL cholesterol (mmol/L)	2.63 ± 0.58	2.78 ± 0.74	2.22 ± 0.75	2.25 ± 0.78	0.017
GOT (U/L)	17.5 ± 5.06	20.1 ± 11.3	16.7 ± 8.99	15.2 ± 8.77	0.075
GPT (U/L)	22.2 ± 10.8	24.6 ± 15.7	24.6 ± 12.3	27 ± 11.5	0.166
Alkaline phosphatase (U/L)	71.7 ± 21	68 ± 19	64.2 ± 21.2	57.2 ± 20.2	0.008
GammaGT (U/L)	15.2 ± 8.01	16.5 ± 6.24	14.2 ± 3.86	14.5 ± 5.01	0.431
Creatinine (mg/dL)	0.72 ± 0.08	0.82 ± 0.13	0.78 ± 0.13	0.78 ± 0.12	0.055
Na (mmol/L)	140.2 ± 2.16	139.6 ± 1.67	139.6 ± 1.14	139.8 ± 1.30	0.824
K (mmol/L)	4.44 ± 0.42	4.32 ± 0.31	4.12 ± 0.27	4.08 ± 0.35	0.063
Calcium (mg/dL)	9.36 ± 0.25	9.68 ± 0.23	9.58 ± 0.24	9.34 ± 0.36	0.246
Phosphorus (mg/dL)	3.28 ± 0.29	3.36 ± 0.34	3.32 ± 0.24	3.14 ± 0.27	0.952
Vitamin D (mcg/L)	21.4 ± 5.223.	27.2 ± 6.53	30.2 ± 7.69	32 ± 14.03	0.015
PTH (ng/L)	52.7 ± 17.5	44.7 ± 14.7	50.5 ± 21.8	38.5 ± 27.7	0.188
C-reactive protein (mg/L)	2.44 ± 1.92	2.34 ± 2.28	1.13 ± 1.02	1.39 ± 1.35	0.682

**Table 3 nutrients-15-04647-t003:** Clinical, hormonal and metabolic parameters in controls at baseline and after 1, 3 and 5 weeks of ketogenic diet.

	Controls (N = 15)				
	Baseline Mean (± SD)	Week 1 Mean (± SD)	Week 3 Mean (± SD)	Week 5 Mean (± SD)	*p*
Clinical parameters					
BMI (Kg/m^2^)	32.8 ± 2.14	27.4 ± 3.14	26.3 ± 2.91	25.5 ± 2.33	0.003
Waist circumference (cm)	96.6 ± 14.2	95.1 ± 14.8	93.2 ± 13.5	92 ± 13.3	0.002
Systolic blood pressure (mmHg)	122.3 ± 8.6	119.4 ± 6.7	115.3 ± 5.6	114.7 ± 6.3	0.025
Diastolic blood pressure (mmHg)	83.2 ± 4.8	81.5 ± 3.6	80.7 ± 4.2	78.2 ± 4.6	0.007
Hormonal parameters					
Fasting serum cortisol (mcg/dL)	11.2 ± 3.44	18.8 ± 1.83	17.9 ± 2.11	15.7 ± 2.05	0.180
Urinary free cortisol (mcg/24 h)	33.4 ± 10.7	29.5 ± 15.7	32.6 ± 17.8	27.9 ± 14.3	0.552
Salivary cortisol	0.76 ± 0.49	0.85 ± 0.59	0.48 ± 0.15	0.52 ± 0.27	0.818
17OHP (mcg/L)	0.32 ± 0.21	0.34 ± 0.28	0.45 ± 0.27	0.41 ± 0.22	0.768
Androstenedione (mcg/dL)	0.71 ± 0.34	1.21 ± 0.75	1.35 ± 1.05	1.15 ± 0.95	0.567
DHEAS (mcg/L)	624.5 ± 324.5	557.1 ± 194.5	587.9 ± 234.5	598.4 ± 205.5	0.768
Cortisone (mcg/L)	18.6 ± 6.5	20.1 ± 5.6	19.5 ± 4.9	18.6 ± 7.3	0.645
Metabolic parameters					
Glycemia (mmol/L)	4.44 ± 0.34	4.03 ± 0.98	4.22 ± 0.71	3.83 ± 0.16	0.074
Insulinemia (microU/mL)	7.46 ± 3.83	10.1 ± 9.51	4.15 ± 3.32	5.65 ± 5.05	0.241
HbA1c (%)	5.14 ± 0.32	5.25 ± 0.63	5.25 ± 0.49	5.11 ± 0.71	0.290
Total cholesterol (mmol/L)	4.97 ± 0.6	4.48 ± 0.2	4.07 ± 0.67	4.07 ± 0.75	0.026
HDL cholesterol (mmol/L)	1.38 ± 0.53	1.47 ± 0.62	1.5 ± 0.69	1.6 ± 0.32	0.001
Triglycerides (mmol/L)	0.82 ± 0.37	0.89 ± 0.27	0.69 ± 0.1	0.71 ± 0.22	0.145
LDL cholesterol (mmol/L)	2.8 ± 0.44	2.69 ± 0.2	2.28 ± 0.1	2.25 ± 0.14	0.132
GOT (U/L)	22.2 ± 10.8	22.5 ± 3.53	22.5 ± 0.71	21.5 ± 2.12	0.212
GPT (U/L)	22.8 ± 6.96	16 ± 2.82	18.5 ± 2.12	18 ± 1.97	0.267
Alkaline phosphatase (U/L)	72.2 ± 19.6	78.5 ± 27.5	69.5 ± 26.1	69 ± 25.4	0.120
GammaGT (U/L)	15 ± 12.1	12.5 ± 7.07	9.5 ± 4.94	9.5 ± 3.53	0.290
Creatinine (mg/dL)	0.71 ± 0.13	0.74 ± 0.11	0.77 ± 0.13	0.76 ± 0.15	0.501
Na (mmol/L)	139.8 ± 1.78	139.5 ± 2.12	138.5 ± 2.09	139 ± 2.02	0.572
K (mmol/L)	4.2 ± 0.44	4.05 ± 0.21	4.15 ± 0.35	3.91 ± 0.28	0.129
Calcium (mg/dL)	9.31 ± 0.26	9.29 ± 0.12	9.45 ± 0.21	9.3 ± 0.27	0.244
Phosphorus (mg/dL)	3.56 ± 0.23	3.55 ± 0.21	3.35 ± 0.51	3.81 ± 0.14	0.753
Vitamin D (mcg/L)	24.6 ± 9.09	34 ± 8.09	37.5 ± 7.78	42.5 ± 17.6	0.514
PTH (ng/L)	43 ± 11.8	32 ± 2.82	37.5 ± 2.12	37 ± 1.41	0.267
C-reactive protein (mg/L)	0.79 ± 0.75	1.42 ± 1.1	1.07 ± 0.65	0.76 ± 0.14	0.443

## Data Availability

Restrictions apply to the availability of some or all data generated or analyzed during this study to preserve patient confidentiality or because they were used under license. The corresponding author will, upon request, detail the restrictions and any conditions under which access to some data may be provided.

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
