# Peer review of "Nutritional Intervention in Cushing’s Disease: The Ketogenic Diet’s Effects on Metabolic Comorbidities and Adrenal Steroids"

_nutrients, 2023, doi:10.3390/nu15214647_

Round 1
Reviewer 1 Report
Comments and Suggestions for Authors
This is an interesting pilot study about the potential value of medical nutrition therapy for the treatment of CD. The protocol is clear; however, the discussion needs to do a better job of connecting the results of your studies with the findings of the highlighted studies. This is necessary to demonstrate what the relationship between the other studies discussed and your study—do they support or contradict your findings? Also, add a paragraph about nutritional considerations of the two test diets. In addition to the potential therapeutic value, what are the potential risks to patients with CD on such a restrictive intake of kcalories? Is it meant to be a long-term intervention? If so, how might patients need to ensure they maintain sound nutritional status? Perhaps these are questions to pose for future research studies on tested dietary intervention?
Line 27: Add Cushing’s Disease in conclusion. Perhaps after “… effective in improving metabolic disorders [insert associated with CD] and ...”
Lines 34-39: The difference between Cushing’s Syndrome and CD needs to be more clearly articulated. Recommend you more clearly distinguish based on recent consensus document: Consensus on diagnosis and management of Cushing's disease: a guideline update. Lancet Diabetes Endocrinol. 2021 Dec;9(12):847-875. doi: 10.1016/S2213-8587(21)00235-7, or use your reference number 20.
Line 36: Define ACTH acronym, Adrenocorticotropic hormone (ACTH)
Line 38: Add “as” after defined. “…defined as Cushings…”
Line 185: Change “The Statistical Packages for Social Science SPSS version 19 (SPSS, Inc.)” to IBM SPSS Statistics Version 19.0. (IBM Corp., Armonk, USA)
Line 192: In results, start with a paragraph that states if all of the participants completed the protocol. Also, add sentence about any reported adverse effects of dietary intervention on participants.
Lines 250 and 255: Change carb to carbohydrate.
Line 255: How does that compare with the ratio of carbs in the two study test diets?
Lines 279-282: Recommend moving into Introduction, put at beginning of Line 56.
Line 303: How do the different pharmacological treatments received by the study participants potentially impact the findings?
Line 315: Add line that notes the need to consider overall nutritional risks for long-term implementation of this dietary pattern.
Comments on the Quality of English LanguageEnglish is good; however, vocabulary selection in parts is pedestrian. Recommend copyedit with focus on ensuring language meets caliber for scientific publications.
Author Response
This is an interesting pilot study about the potential value of medical nutrition therapy for the treatment of CD. The protocol is clear; however, the discussion needs to do a better job of connecting the results of your studies with the findings of the highlighted studies. This is necessary to demonstrate what the relationship between the other studies discussed and your study—do they support or contradict your findings? Also, add a paragraph about nutritional considerations of the two test diets. In addition to the potential therapeutic value, what are the potential risks to patients with CD on such a restrictive intake of kcalories? Is it meant to be a long-term intervention? If so, how might patients need to ensure they maintain sound nutritional status? Perhaps these are questions to pose for future research studies on tested dietary intervention?
Dear reviewer,
Thanks for taking time to review our paper and providing comments and suggestions for changes that improved the manuscript.
We agree with you comments. As you suggested we tried to compare the results of the current study with the studies conducted by Stimson et al. as reported in lines 283-296.
We also added a paragraph with the nutritional percentages of macronutrients for VCLKD and LCKD.
With regard to the risks of a restrictive intake of kcalories in patietns with CD, we added it as a limitation of the study, because in our opinion the short duration of the protocol diet could not significantly impact on health in these patients. We hypothesize a maximum duration of ketogenic protocol of about 2 months.
Line 27: Add Cushing’s Disease in conclusion. Perhaps after “… effective in improving metabolic disorders [insert associated with CD] and ...”
Thanks for the comment. We added as you suggested.
Lines 34-39: The difference between Cushing’s Syndrome and CD needs to be more clearly articulated. Recommend you more clearly distinguish based on recent consensus document: Consensus on diagnosis and management of Cushing's disease: a guideline update. Lancet Diabetes Endocrinol. 2021 Dec;9(12):847-875. doi: 10.1016/S2213-8587(21)00235-7, or use your reference number 20.
Thanks for the comment. We clearly distinguished between Cushing's syndrome and Cushing's disease and added the reference you suggested.
Line 36: Define ACTH acronym, Adrenocorticotropic hormone (ACTH)
Thanks for the comment. We added the definition as you suggested.
Line 38: Add “as” after defined. “…defined as Cushings…”
Thanks for the comment. We added "as".
Line 185: Change “The Statistical Packages for Social Science SPSS version 19 (SPSS, Inc.)” to IBM SPSS Statistics Version 19.0. (IBM Corp., Armonk, USA)
Thanks for the comment. We changed the sentence as you kindly suggested.
Line 192: In results, start with a paragraph that states if all of the participants completed the protocol. Also, add sentence about any reported adverse effects of dietary intervention on participants.
Thanks for the comment. We added the information about adverse effects at line 199.
Lines 250 and 255: Change carb to carbohydrate.
Thanks for your suggestion. We changed it.
Line 255: How does that compare with the ratio of carbs in the two study test diets?
Thanks for the comment. We added a personal comment in the text at line 285-295
Lines 279-282: Recommend moving into Introduction, put at beginning of Line 56.
Thanks for the suggestion. We moved at line 60.
Line 303: How do the different pharmacological treatments received by the study participants potentially impact the findings?
Thanks for the question. We tried to comment it at line 297.
Line 315: Add line that notes the need to consider overall nutritional risks for long-term implementation of this dietary pattern.
Thanks for the comment. We added it as a limitation of the study.
Reviewer 2 Report
Comments and Suggestions for Authors
Nutritional intervention in Cushing’s disease: the ketogenic 2
diet effects on metabolic comorbidities and adrenal steroids
A very low-calorie ketogenic diet (VLCKD) is characterized by low daily caloric intake , low carbohydrate intake and normoproteic contents which induces a significant weight loss and an improvement in lipid parameters, blood pressure, glycaemic indices and insulin sensitivity in patients with obesity and type 2 diabetes mellitus. Cushing’s syndrome (CS) is caused by an endogenous or exogenous excess of glucocorticoids and leads to many comorbidities including cardiovascular disease, obesity, type 2 diabetes mellitus and lipid disorders. The aim of this study is to verify if the VLCKD could be used for the treatment of CS comorbidities VLCKD for the treatment of CS comorbidities, analysing different parameters. Study design, methods are adequately described, tables and figures too. References must be up-dated
(similar article:Guarnotta V, Emanuele F, Amodei R, Giordano C. Very Low-Calorie Ketogenic Diet: A Potential Application in the Treatment of Hypercortisolism Comorbidities. Nutrients. 2022 Jun 9;14(12):2388. doi: 10.3390/nu14122388)
Author Response
A very low-calorie ketogenic diet (VLCKD) is characterized by low daily caloric intake , low carbohydrate intake and normoproteic contents which induces a significant weight loss and an improvement in lipid parameters, blood pressure, glycaemic indices and insulin sensitivity in patients with obesity and type 2 diabetes mellitus. Cushing’s syndrome (CS) is caused by an endogenous or exogenous excess of glucocorticoids and leads to many comorbidities including cardiovascular disease, obesity, type 2 diabetes mellitus and lipid disorders. The aim of this study is to verify if the VLCKD could be used for the treatment of CS comorbidities VLCKD for the treatment of CS comorbidities, analysing different parameters. Study design, methods are adequately described, tables and figures too. References must be up-dated
(similar article:Guarnotta V, Emanuele F, Amodei R, Giordano C. Very Low-Calorie Ketogenic Diet: A Potential Application in the Treatment of Hypercortisolism Comorbidities. Nutrients. 2022 Jun 9;14(12):2388. doi: 10.3390/nu14122388)
Dear reviewer,
Thanks for taking time to review our paper and providing comments and suggestions for changes that improved the manuscript.
We very appreciate your work.
The paper you cited is a speculative review which aimed to evaluate the potential beneficial effects of VLCKD in Cushing's syndrome.
This current article is the direct experience of the use of ketogenic diet in patients with Cushing's disease compared to healthy controls. We are conscious on the need of perform further larger studies to confirm these preliminary data.